# Adopting yield-improving practices to meet maize demand in Sub-Saharan Africa without cropland expansion

Fernando Aramburu-Merlos [1,2], Fatima A. M. Tenorio[1], Nester Mashingaidze[3], Alex Sananka[3], Stephen Aston[3], Jonathan J. Ojeda[4] & Patricio Grassini [1] ✉

Maize demand in Sub-Saharan Africa is expected to increase 2.3 times during the next 30 years driven by demographic and dietary changes. Over the past two decades, the area cropped with maize has expanded by 17 million hectares in the region, with limited yield increase. Following this trend could potentially result in further maize cropland expansion and the need for imports to satisfy domestic demand. Here, we use data collected from 14,773 smallholder fields in the region to identify agronomic practices that can improve farm yield gains. We find that agronomic practices related to cultivar selection, and nutrient, pest, and crop management can double on-farm yields and provide an additional 82 million tons of maize within current cropped area. Research and development investments should be oriented towards agricultural practices with proven capacity to raise maize yields in the region.

The beginning of the new millennium has seen fast expansion of the area cropped with maize in Sub-Saharan Africa (SSA), which has become several times larger than that for other crops in the region and equivalent to that in the U.S. Corn Belt[1] (Fig. 1a). The sharp increase in maize demand between now and year 2050 (+233%)[2] will add further pressure on expanding maize area or require costly imports, which is a concern given the limited monetary reserves of most countries in the region and recent shocks in commodity prices[3]. A substantial body of evidence shows that the average maize yield (2 tons per hectare) is five times less than the yield potential as determined by the climate and soils that prevail in SSA maize-producing areas[4,5]. Hence, an opportunity exists for SSA to produce substantially more maize on current cropland by narrowing the existing yield gap. Such an approach can help avoid maize imports and alleviate pressure to expand cropland at the expense of natural ecosystems and the cultivation of marginal lands.

Yield improvement has remained elusive for SSA maize systems despite the investments in agricultural research and development (AR&D) programs made by African governments, international donors, and charitable foundations[6]. For instance, the current rate of maize yield gain in SSA is four to five times slower than in Southeast Asia and South America (Fig. 1b). Quantifying the impact of agronomic technologies on farmer yields across a large geographic area is difficult given the multitude of variables that can influence yield and their interactions with climate and soils[7]. Previous studies in SSA have relied on a relatively small number of field surveys, typically exploring a narrow range of management practices and environments, and/or field trials focusing on individual practices[8–12]. Analysis of large farmers' field databases, complemented with fine-resolution climate, soil, and terrain information, can help identify suites of management practices that consistently lead to higher yield in a given environment[13,14]. In turn, this approach can help prioritize investments in AR&D programs towards those practices that are more effective at increasing farmer yield and orient policy to facilitate their adoption across different socio-economic contexts[15].

Identification of yield-improving management practices in maize systems in SSA can re-orient researchers, policymakers, and donors toward solutions for increasing maize production, without large maize area expansion or reliance on costly imports. Here we identify management practices with the potential to increase on-farm yields in

[1]Department of Agronomy and Horticulture, University of Nebraska-Lincoln, Lincoln, NE, USA. [2]Instituto de Innovación para la Producción Agropecuaria y el Desarrollo Sostenible (IPADS) Balcarce (INTA-CONICET), Balcarce, Buenos Aires, Argentina. [3]One Acre Fund, Nairobi, Kenya. [4]Regrow Ag, Brisbane, QLD 4075, Australia. ✉e-mail: pgrassini2@unl.edu

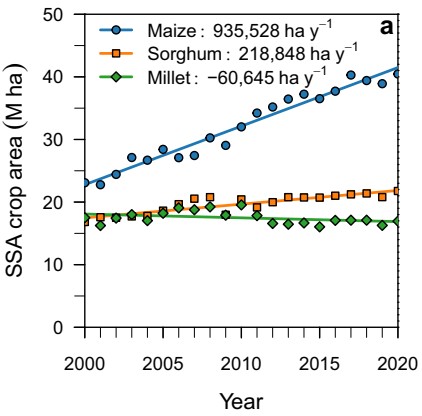

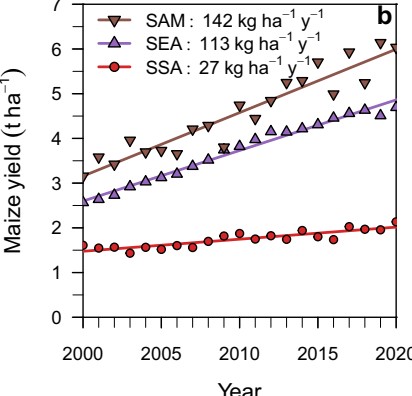

**Fig. 1 | Cereal area and maize yield trends.** Trends in **a** harvested area of maize, sorghum, and millet in Sub-Saharan Africa (SSA) and **b** average maize yield in SSA, Southeast Asia (SEA), and South America (SAM) between 2000 and 2020. Data from FAOSTAT[1]. Values in each panel show the rate of **a** cropland expansion and **b** yield gain for each crop or region. Source data are provided as a Source Data file.

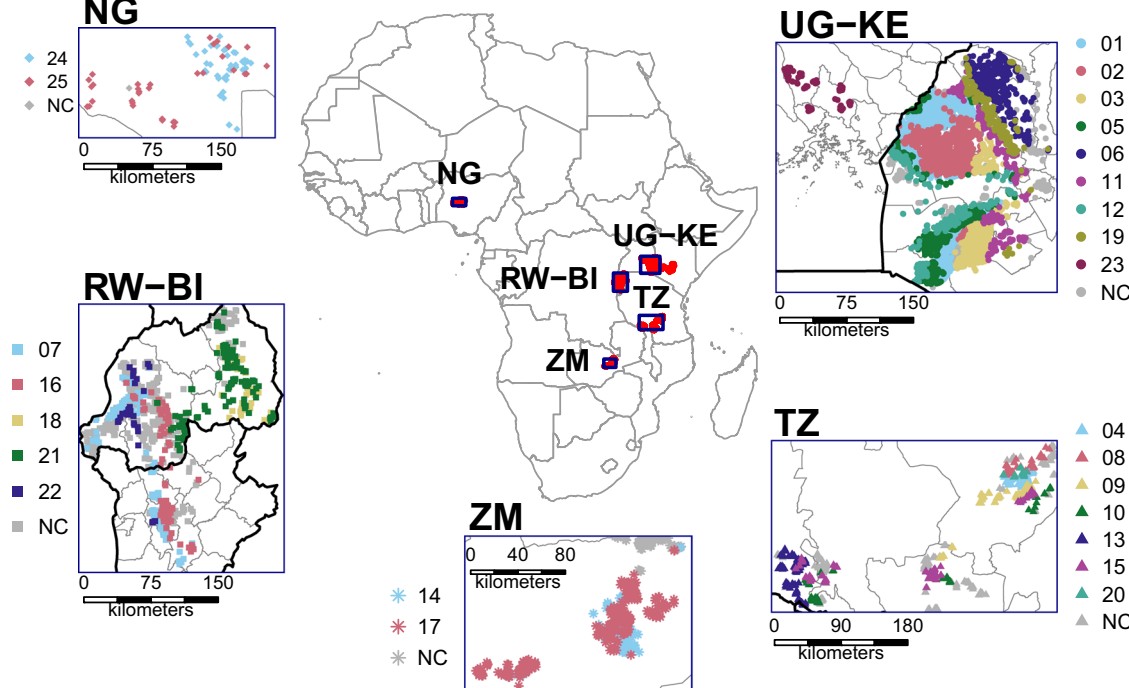

**Fig. 2 | Location and stratification of maize fields.** Location of the 14,773 maize fields included in the field-level farmer database in five regions of Sub-Saharan Africa: Nigeria (NG), Rwanda-Burundi (RW-BI), Zambia (ZM), Uganda-Kenya (UG-KE), and Tanzania (TZ). Colors represent different climate zones (CZ; $n$ = 25) as determined by growing-degree days and aridity index. A separate conditional inference tree analysis was performed for each climate zone (identification number is shown in map legends). Gray points (NC) are fields that fall in a climate zone but with insufficient observations for climate zone-specific yield analysis. Source data are provided as a Source Data file.

different maize-producing areas in SSA. We show that maize yields in SSA can double by following improved crop and soil management practices, and we then discuss implications for AR&D prioritization and policy.

## Results

We analyzed field-level yield and management data collected from 14,773 smallholder fields in SSA (Fig. 2). Analyzing large farmer databases is challenging given the variation in climate, soil, and management practices across fields and the lack of formal experimental design. Our database in SSA was not an exception, as it included fields located in humid and drier environments and following contrasting management practices (Figs. S1 and S2). Here we used advanced statistical methods, coupled with spatial databases of climate, soil, and terrain, to quantify the impact of management practices and their interactions on farmer yields across a wide range of maize-producing systems with varying environmental conditions. To avoid the confounding effect of spatial and temporal variation in climate and soil, we stratified maize fields based on climate zones and performed a separate analysis for each of them, including soil, terrain, and in-season weather variables to account for the residual variation within climate zones. In total, we considered 25 production environments with contrasting climate conditions located in (i) north-central Nigeria (NG), (ii) Rwanda and Burundi (RW-BI), (iii) central Zambia (ZM), (iv) southwestern Tanzania (TZ), and (v) eastern Uganda and western Kenya (UG-KE) (Fig. 2). Range in climate and soil across the fields included in our

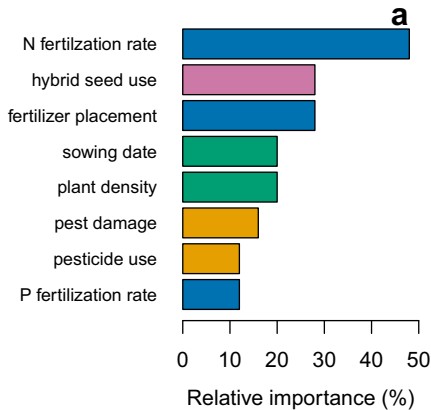
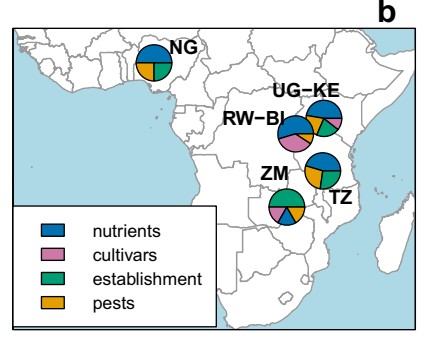

**Fig. 3 | Agronomic practices explaining variation in maize yield in Sub-Saharan Africa. a** Relative importance was calculated as the proportion of climate zones where an agronomic practice had a significant effect on maize yield, as identified in the conditional inference tree analysis (Fig. S12). Colors represent different types of practices (nutrients, cultivars, establishment, or pest management). N refers to nitrogen and P to phosphorus. **b** Pie-charts show the relative importance of each type of agronomic practice at explaining yield variation in each maize-producing region: Nigeria (NG), Uganda & Kenya (UG-KE), Rwanda & Burundi (RW-BI), Tanzania (TZ), and Zambia (ZM). We could not evaluate the hybrid effect in NG and TZ because most fields were planted with hybrids. Source data are provided as a Source Data file.

database portrayed well that across the maize producing area in SSA, making our analysis relevant for the entire region (Figs. S3 and S4).

Our analysis based on the stratification of maize fields by climate zones and conditional inference tree models revealed several agronomic practices that explained variation in yield among farmer fields across production environments (Fig. 3; Fig. S12). These practices included nutrient rate and placement, use of hybrids, sowing date, weed and pest control, and plant density. Higher yields were associated with higher rates of nitrogen (N) and phosphorus (P) fertilizer, proper fertilizer placement (in a hole instead of surface application), use of hybrids instead of open-pollinated varieties, early sowing dates, higher plant densities, and effective pest control. We also detected synergistic effects among these practices. For example, the yield benefit associated with hybrids was largest when the crop was sowed early, and N fertilizer rates were high (e.g., climate zone #2 in Fig. S12). Conversely, we could not detect any impact of hybrid traits on on-farm yields (e.g., crop cycle duration, year of release, and disease tolerance). We suspect that poor agronomic management overrides the impact of genetic traits on yield. An additional analysis based on machine learning confirmed the positive impact of these agronomic practices and provided insights into their individual response functions and other factors that can be tuned to further increase yields such as weeding (Fig. 4).

At question is how much room exists to raise the current yield through agronomic management. Previous works based on robust crop modeling have estimated average yield potential of 10.6 t ha$^{-1}$ for rainfed maize across our study regions based on local weather and soil data[4,5] (Fig. 5). Thus, given the regional average yield of 1.7 t ha$^{-1}$, the resulting yield gap of 8.9 t ha$^{-1}$ suggests ample room to increase maize yields through improved agronomic management. However, complete closure of the yield gap requires copious amounts of inputs and a high degree of sophistication in crop management and soil practices to avoid any yield limitation or reduction. Hence, a more realistic goal to maximize the return to inputs and labor is to target a certain fraction of the yield potential. For instance, measured yields in well-managed crops in local research stations in our study area reached 7.5 t ha$^{-1}$, which represents 70% of the yield potential. Such a level of yield gap closure is consistent with that achieved by farmers with full access to input, markets, and extension services in other regions of the world[16,17].

Our analysis of farmer data allowed us to identify suites of management practices that lead to consistent yield improvement across

environments (Fig. 5). For this assessment, we estimated the average yield of maize fields adopting contrasting technological levels across regions, after accounting for differences in the environmental background, based on the means of a linear mixed-effect model with technology level as a fixed effect and climate zone and year as random effects. The lowest technological level was defined as that with low N fertilizer rates and plant densities, open pollination varieties, and average or late sowing dates. Farmers following the lowest technological level got an average yield of 1.8 t ha$^{-1}$, which is similar to the average maize yield in the region and SSA[1]. Thus, this yield level can be taken as a baseline to assess the impact of improving management practices on farmer yields. For example, farmers who had sowed hybrids *and* applied relatively high N fertilizer rates (*ca.* 50 kg ha$^{-1}$) attained yields that were 61% higher than the baseline (Fig. 5). An additional yield increase was apparent for farmers who had *also* sowed earlier and increased plant density, with a resulting yield of 4.3 t ha$^{-1}$, which, in turn, was 2.4 times higher than the baseline yield. Thus, the adoption of improved management practices can narrow the yield gap by *ca.* 30%, which is equivalent to 2.5 t ha$^{-1}$.

In principle, one would expect our findings to be applicable to other maize-producing regions in SSA given the similarity in soil and climate conditions between them and our study area (Figs. S3 and S4). Thus, we extrapolated the relative yield gap closure derived from the adoption of improved agronomic practices in our study area as shown in Fig. 3 (*i.e.*, use of hybrid, higher N and density, and early sowing) to the entire maize area in SSA (see "Methods" section and Supplementary Note 2.1). Our extrapolation approach is valid as inferred from validation of the results using a spatial machine-learning model that accounts for potential differences in climate and soils affecting yield responses to management practices (Supplementary Note 2.2 and Fig. S5). Our analysis showed that adoption of improved agronomic management at the regional level would increase current SSA maize output from 80 million tons to 168 million tons on existing maize area (Table 1). This scenario will allow the region to come near self-sufficiency for maize by the year 2050, drastically reducing land and import requirements. On the other hand, if current rates of yield improvement persist in the future, the region will not be able to meet domestic demand on existing cropland, requiring an additional 28 million hectares of land or 76 million tons of maize imports to meet the demand. In turn, pressure on cropland expansion will put at risk natural ecosystems and drive crop production into marginal environments.

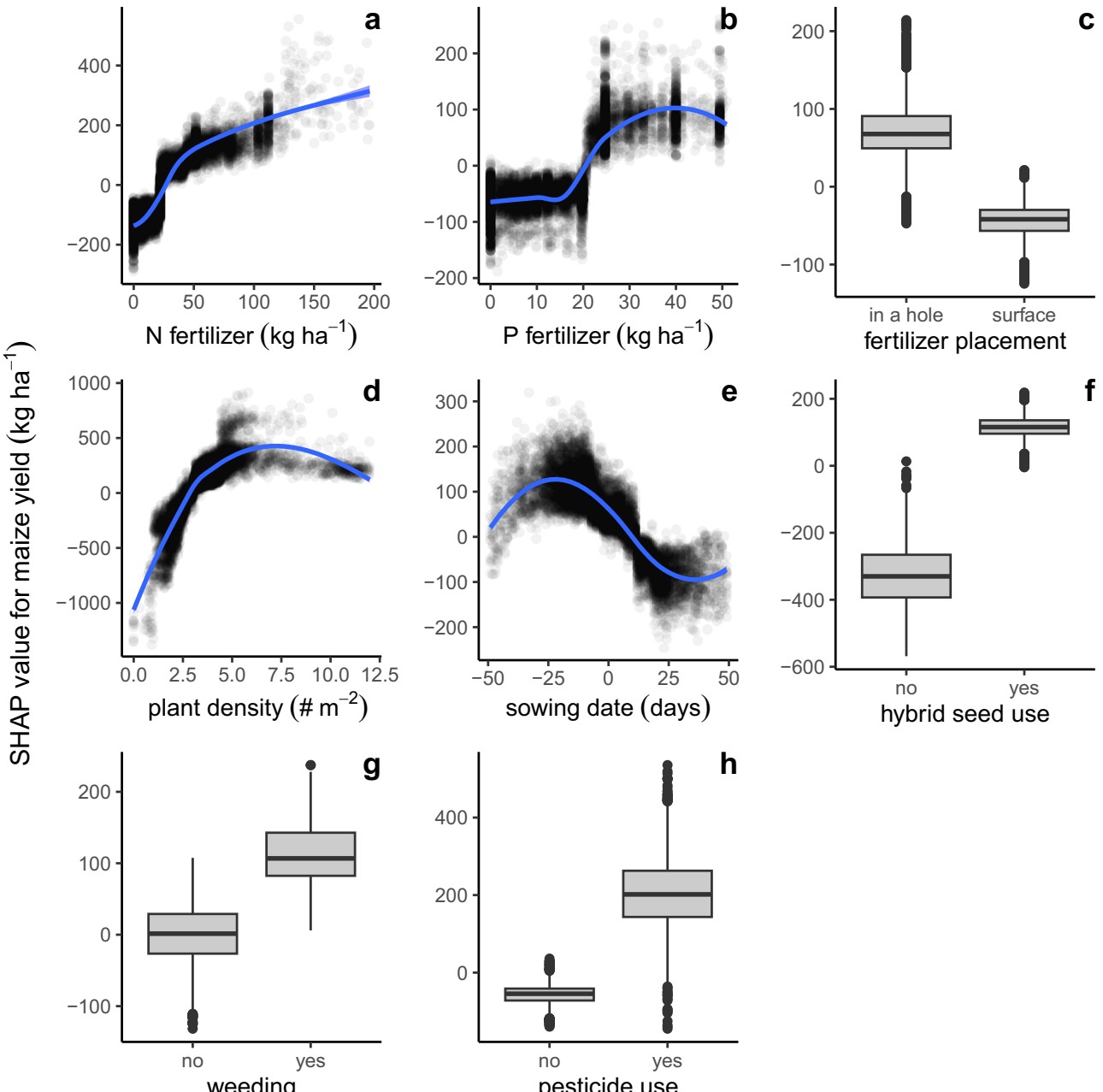

**Fig. 4 | Effect of individual crop management practices on maize yields in Sub-Saharan Africa based on a machine-learning model.** Shapley additive explanation (SHAP) values for the nine agronomic practices with the greatest impact on maize yields. SHAP values were computed based on a Gradient Boosting Machine model trained with the 14,773 maize fields included in the field-level farmer database and biophysical covariables listed in Table S2. Each SHAP value shows how much each agronomic practice contributes to the maize yield variation across smallholder fields as predicted by the machine-learning model. In (**a**), (**b**), (**d**), and (**e**), blue lines were fitted with LOESS regressions. In (**c**), (**f**–**h**), the central line of each boxplot indicates the median, the box encompasses the interquartile range, and the whiskers extend up to 1.5 times the interquartile range while more extreme values are depicted individually as outliers. All maize fields ($n = 14,773$) are included in (**a**–**c**) whereas (**d**) includes 10,166 fields, (**e**) 11,179, (**f**) 13,543, (**g**) 10,967, and (**h**) 9,088 due to missing information for some maize fields. N refers to nitrogen and P to phosphorus. Source data are provided as a Source Data file.

## Discussion

The positive impact of improved agronomic practices on maize yield in SSA has long been known. Reports back from the 1960s show results that are not too different from the ones shown here[18–20]. However, there has been an incessant call in more recent decades for SSA cropping systems to embrace a range of approaches of diverse ideology, focus, and labeling, including conservation agriculture[21,22], agroecology[23,24], climate-smart practices[25,26], regenerative agriculture[23,24,27], nature-based solutions[28,29], and digital

agriculture[30–32]. These approaches are promoted based on the assumption that they will contribute to a wide range of societal goals associated with food security, climate change, and the conservation of natural resources. A concern is that their labeling, broad goals, and (sometimes) detachment from the local context can shift the focus away from the urgent need to increase crop yields in SSA and the most obvious agronomic solutions to achieve this goal within a reasonable timeline. Here we showed that farmers in SSA could benefit from an explicit effort from national and international programs to facilitate

the adoption of improved agronomic practices in relation to soil fertility, cultivar selection, sowing date, plant density, and pest control. We show that maize yields in the region double when farmers adopt improved agronomic practices. In turn, increasing average yield can help increase local and regional food security and improve smallholders' income[33,34]. Indeed, when our results were scaled out over the entire maize area in SSA, we found that agronomic improvement can drastically reduce the need for cropland expansion and/or imports. Failure to narrow the yield gap substantially can have severe negative socio-economic and environmental consequences. Therefore, the region should not be used as the testing ground to promote approaches that have not been empirically validated in their capacity to increase on-farm yields.

While it may be argued that our results rely on observational data and predictive models rather than field experiments, limiting our capacity to establish causal relationships, our findings are supported by thousands of field observations across major agro-ecological zones

where maize is grown in SSA, a robust spatial framework, and advanced statistical and machine-learning methods. Furthermore, our work overcomes major limitations from previous studies assessing yield constraints in SSA. First, many previous analyses have relied on farmer surveys, which is problematic because farmers frequently overestimate yields, particularly in small plots[8,35–39]. In contrast, our study is based on direct yield measurements derived from crop samples collected following consistent protocols. Second, previous yield gap analyses in SSA are narrow in geographic scope and range of agronomic practices[8–10,35,40,41]. In contrast, our study includes thousands of farmers from seven countries, across 25 production environments, including farmers with access to inputs which, in turn, extend the range of agronomic practices in our database, increasing the odds to identify yield-improving practices[42]. Finally, the combination of a climate spatial framework, and statistical and machine-learning methods, together with soil and terrain databases, allowed us to control for the confounding effects of varying climate, soil, and terrain factors in relation to the impact of agronomic practices on on-farm yields. Application of the same approach over other geographies and crop systems could help identify, with a modest investment and in a relatively short timeframe, key yield constraints for all major cropping systems in SSA, providing a basis for researchers, extension workers, and policymakers to prioritize their efforts.

Reaching maize self-sufficiency in SSA is not 'mission impossible' but rather requires strong investment in AR&D and proper policy to focus on those technologies with the greatest probability to deliver yield gains within a short time and at a relevant spatial scale. Our study shows that those technologies already exist, for example, the use of hybrids combined with proper nutrient, pest, and crop establishment practices. While it is unrealistic to imagine all SSA farmers adopting these practices immediately, the period between now and the year 2050, which coincides with the sharp increase in maize demand in SSA, offers a reasonable window of time to promote improved agronomic practices and the means for their adoption, especially if environments with high and more stable yield potential and reasonable access to inputs and markets are prioritized. In turn, reaching an average yield of 4.2 t ha$^{-1}$ by the year 2050, equivalent to a yield gap closure of 30%, would require annual rates of yield gains for maize that are *ca.* three times compared to current ones and comparable to those observed in other tropical and subtropical rainfed maize producing regions in South America and Southeast Asia. Maize yields nearly doubled in those two regions over the past two decades, and the same scenario is possible for SSA through the adoption of improved agronomic management. Moreover, we note that while adopting yield-improving practices has the potential to narrow *ca.* one-third of the yield gap, there would still be another two-thirds that remains, suggesting that further yield improvements are possible but would require more inputs and fine-tuning other management practices and interactions. For instance, our study shows that doubling the fertilizer rate relative to the high-technology farmers, together with proper hybrids, crop stand, and weed control resulted in on-station yields that reached 70% of yield potential.

Our study is subjected to several limitations and uncertainties. First, our scenario assessment relies on extrapolating the results from

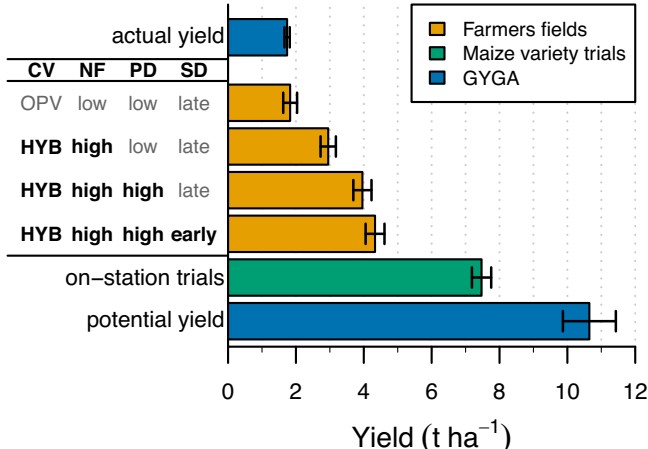

**Fig. 5 | Maize yield in Sub-Saharan Africa as influenced by adoption of management practices.** Orange bars show maize yields of groups of fields across climate zones following combinations of cultivar (CV), nitrogen fertilizer (NF), plant density (PD), and sowing date (SD). Each practice has two categories: open pollination varieties (OPV) or hybrid (HYB) for cultivar; low (<14 kg ha$^{-1}$) or high (>42 kg ha$^{-1}$) N fertilizer; low (<3 plants m$^{-2}$) or high (>4 plants m$^{-2}$) plant density; and late (equal or later than climate zone average) or early (>9 days earlier than climate zone average) sowing date. Fields were grouped based on the adoption of management practices and yield differences between groups were evaluated with a linear mixed-effect model with technology level as fixed effect and climate zone and year as random effects to account for variation in biophysical background. Data are presented as marginal mean yields +/- standard error of the mean. The number of farmer fields included in each group was 82 for the baseline, 314 for hybrid + high NF, 285 for hybrid + high NF + high PD, and 243 for hybrid + high NF + high PD + early PD. Also shown as reference are average actual and potential yields across 25 climate zones (GYGA-Global Yield Gap Atlas; www.yieldgap.org) and measured average yields in well-managed maize variety trials (148 plots from three sites, three years, and 30 cultivars) with their respective standard errors. Source data are provided as a Source Data file.

## Table 1 | Scenario assessment of maize self-sufficiency by 2050

| Scenario | Average yield (t ha$^{-1}$) | Production (million t) | Demand (million t) | SSR | Balance (million t) | Extra land requirement (million ha) |
|---|---|---|---|---|---|---|
| Current situation | 2 | 80 | 79 | 1.0 | +1 | 0 |
| Same rates of yield gain | 2.7 | 108 | 184 | 0.6 | −76 | 28 |
| Acceleration of yield gain | 4.2 | 168 | 184 | 0.9 | −16 | 4 |

Average maize yield, production, demand, self-sufficiency ratio (SSR), and balance in Sub-Saharan Africa (SSA) for the current situation (around the year 2020) and for two scenarios of yield improvement between 2020 and 2050: same rate of yield gain as in the last 20 years and acceleration of yield gain to reach the yield gap closure level achieved by farmers adopting improved management practices in our dataset. The SSR was estimated as the ratio between maize production and demand. All scenarios assumed no change in the current maize area in SSA. The extra land requirement to meet maize demand by 2050 is also shown. See "Methods" section and Supplementary Notes 2 for details on calculations.

our analysis to the maize-growing area in SSA. However, similarity in climate and yield between our study region and other maize-producing areas in SSA, consistency between the results from the conditional regression trees and those from a spatial machine-learning approach accounting for differences in climate and soil, and previous evidence on the widespread positive impact of our identified yield-improving management practices[7,43–45] give confidence on our regional extrapolation. Second, despite measuring yields on every field represents a major improvement regarding previous studies, our data still suffer from uncertainties, which is typical of farmer surveys in developing countries. However, considering the large size of our database (14,773 observations) and the consistency of results across 25 production environments, we believe our results are robust. Finally, our pledge to promote improved agronomic practices in SSA should not be taken as a prescription. Indeed, local tuning in each environment is needed to make recommendations that account for gradients in soil fertility resulting from differences in field history and diversity in farming systems[7,46]. Still, the overall message about the importance of agronomic management as a pathway to increase crop yield and improve food security in SSA needs to be clear for international donors and national programs and our study makes an important contribution to document the potential on-farm impact.

Failure to achieve the required level of intensification in maize systems in SSA can have far-reaching consequences, leading to a substantial increase in maize imports and/or cropland expansion (Table 1). Indeed, these results are conservative since our scenario assessment assumed that the new cropland has the same productivity as the current area, which may be too optimistic as cropland expansion may occur in less productive lands[47,48]. Likewise, our assessment assumes no negative impact of climate change on maize yields, which may add further difficulty to improving yields in the region[49,50]. Ultimately, lower yields due to cropland expansion into marginal land and climate change will add further pressure to intensify maize production on existing cropland to reach a reasonable level of self-sufficiency while avoiding large conversion of new land in SSA for agriculture, reinforcing the overall message from our study about the importance of re-orienting AR&D programs and policy to those key management factors likely to benefit farmers and facilitate farmers' access to inputs and technical information. Likewise, achieving the desired level of intensification will require strong policy interventions to overcome major smallholder financial challenges by facilitating them with financial resources, credit services, agricultural insurance, and market access. For example, low fertilizer application in SSA is partially a consequence of high fertilizer prices (relative to the grain price) and low economic incentives to intensify crop production, which, in turn, results from a large uncertainty on the return of investment from agriculture and high rural population dependency on off-farm income[51–53]. Illustratively, we estimated that the cost of implementing the yield-improving practices identified in our study (hybrid seed, early planting, proper density, and higher use of N and P fertilizer) ranged from US$200 to US$250 per hectare depending upon country and based on 2023 prices. Hence, yield improvement will surely require more than improved crop management but also proper markets, infrastructure, institutions, and policies to lower fertilizer and seed costs and facilitate their adoption by farmers. Ultimately, an area greater than the US Corn Belt is waiting for yield intensification in SSA, but no intensification can be achieved without agronomic improvement.

## Methods
### Ethics compliance
The Research Compliance Services at the University of Nebraska-Lincoln determined that this project does not meet the definitions of human subjects research under regulatory requirements at 45 CFR 46.102 and hence does not require IRB approval. The information requested from the farmers was about the farm and not about the individual. Under U.S. regulation, this is not considered to be human subjects research. The Office of Human Research Protections (OHRP) International Compilation of Human Research Standards was also referred to in making this decision since the project was conducted outside of the U.S. Participants were informed that their participation was voluntary. Please see Supplementary Note 1 for the survey questionnaire.

### Database description
We used crop yields and management practices data collected from smallholders' maize fields from 2016 to 2022. All fields corresponded to maize grown in pure stands (no intercropping). Data were collected from five maize-producing regions in SSA: (i) north-central Nigeria (NG, $n = 115$), (ii) Rwanda and Burundi (RW-BI, $n = 2720$), (iii) central Zambia (ZM, $n = 861$), (iv) southwest Tanzania (TZ, $n = 3710$), and (v) eastern Uganda and western Kenya (UG-KE, $n = 7367$) (Fig. 2). There is one crop season per year in TZ, NG, and ZM because of water limitation during the rest of the year (Fig. S1). In contrast, in UG-KE and RW-BI, farmers can grow rainfed crops in two seasons in the same year. For simplicity, we referred to these seasons as first and second based on their chronological order after the end of the dry season. Our database included data from the second season only in the case of RW-BI (Fig. S2).

Data were collected by "One Acre Fund" (https://oneacrefund. org/), an NGO that provides smallholder farmers access to agricultural training, credit, crop insurance services, and farming supplies. About half of the fields in the database comprised farmers who subscribed to the One Acre Fund program whereas the other half did not. We see the inclusion of farmers with varying levels of technology adoption in the database as an advantage as it allows to increase the variation in management practices across farmer fields[42]. Because farmers engaged with the One Acre Fund program have greater technology adoption, it was not surprising that the average maize yield of our database (3 t ha$^{-1}$) was higher than the average maize yield in those same regions (1.7 t ha$^{-1}$)[1].

Maize grain yield, plant density, and row spacing were measured in two randomly placed boxes of 36 m$^2$ at harvest, avoiding field edges. Field geolocation was recorded in 70% of the observations. When missing, the field geolocation was defined based on the nearby town (21%) or associated district (9%) location for the purpose of retrieving climate data. Management practices associated with each field were reported by farmers, including sowing and harvest dates, cultivar name, fertilizer inputs (types and total quantities for both organic and inorganic), fertilization method, liming, weeding, and pesticides (mainly insecticides to control fall armyworms). Farmers also reported the incidence of adversities (such as pests, diseases, Striga witchweed, hail, and excess water). Field size was reported by farmers and, in those cases in which farmers could not provide an accurate measure of their field size, or there was a strong indication of mistakes (*e.g.*, nutrient fertilizer rates out of range), One Acre Fund personnel took in-situ measurements to determine field size. Input rates per hectare were calculated as the ratio of the farmer-reported input amount and field size. Data were subjected to quality control to remove unlikely values. Maize yield outliers were detected with a Bonferroni Outlier Test, after which only four observations were removed. Observations with plant densities and fertilizer rates higher than four standard deviations from the mean were excluded (cutoffs: 12 plants m$^{-2}$ and 400 kg ha$^{-1}$ of fertilizer) as well as those without geolocation ($n = 351$), N or P data ($n = 520$) and atypical sowing dates ($n = 42$). After quality control, the database contained a total of 14,773 field observations.

The management practices variables used in the statistical analysis are described in Table S1. Inorganic fertilizer rates were converted to nutrient rates (in elemental nutrients) following typical fertilizer nutrient contents. Organic fertilizers were encoded separately in two

binary variables and one continuous variable, indicating whether compost was used, if that compost contained manure, and compost application rate. Likewise, cultivars were classified into hybrids or open-pollination varieties, which included local varieties, retained seeds, and improved open-pollination varieties. For hybrids, we retrieved the associated crop cycle maturity (short, medium, and long), disease tolerance traits, and year of release from companies' seed catalogs. The reported incidence of diseases and insect pests (*e.g.*, anthracnose, aphids, blight, cutworms, drought, fall armyworm, stemborer, termites, and stalk or kernel rot) were simplified to two binary variables indicating whether the crop was affected by pests and/or diseases. Infestation by parasitic witchweeds (*Striga hermonthica and S. asiatica*) was considered as a separate variable. Fertilization methods were also simplified to whether the fertilizer was applied inside a hole or broadcasted on the surface. Number of weeding operations was simplified to zero, one or two or more weeding per season. Sowing dates were expressed as a deviation from the estimated average sowing date for each climate zone-season combination.

## Grouping fields according to biophysical background

Landscape heterogeneity and the unavailability of accurate weather and soil data can introduce uncertainty to the analysis and/or lead to confounding effects. To manage these issues while retaining enough statistical power, we first clustered the observations based on climatic conditions and distance; and second, we included elevation, weather, soil, and topography properties in the statistical analysis as covariables (Table S2).

Fields were grouped based on their location using the climate zones scheme developed by the Global Yield Gap Atlas Project (GYGA)[54]. Despite the underlying uncertainty of global gridded databases[55,56] and remaining climatic and soil type variation within climate zones (Fig. S10), previous studies have shown that stratifying fields by climate zone is a robust approach to group fields with similar biophysical backgrounds and analyze the impact of management practices on farmer yields while minimizing confounding effects[13,14,57–59]. To better capture the sharp climatic gradients in some production environments (*e.g.*, mountainous areas in Rwanda and Kenya), we re-delineated the original climate zones to increase their spatial resolution from 5 arc-minutes to 30 arc-seconds (*ca.* 1 km). For that purpose, we derived the three bioclimatic variables used in the climate zone framework from WorldClim2[56]: annual growing-degree days (GDD), aridity index (AI), and temperature seasonality (TS). Then, fields were grouped into climate zones while removing isolated observations by keeping those sites that were less than three standard deviations from the median distance across sites within the climate zone. In the case of climate zones with two maize seasons, each crop season was considered as a separate climate zone. To ensure enough statistical power[60], we considered those climate zones that contained more than 200 fields. For simplicity, we showed here the aggregated results, while results for the individual climate zones are shown in Fig. S12.

Our statistical models included biophysical properties as covariables to account for any residual spatial variation in climate and soil that was not captured by our climate zones, (Fig. S10). These covariables included weather, soil, topography, and altitude variables (Table S2). Field elevation, which was retrieved from the Amazon Web Services Terrain Tiles[61], is strongly correlated with variation in temperature. Likewise, to account for inter-annual weather variation, we calculated total precipitation during the growing season as well as for three crop phases: early, flowering, and grain filling. To do this, we defined the growing season for each climate zone-year combination as the period between 10% of sowing progress and 50% of harvest progress. Daily precipitation data was retrieved from CHIRP[62]. In addition, for each observation with field-level coordinates data, we retrieved the root-zone plant-available water-holding capacity from the World Soil Information database[63], and soil clay content, pH, organic carbon, and

effective cation exchange capacity from iSDA[64]. Lastly, we calculated the topography wetness index (TWI) from the elevation data[65,66]. The latter accounts for differences in slope and terrain that influence the soil water balance and erosion. High TWI values are associated with flat terrain and deep soils, whereas small values indicate steep slopes, greater risk of erosion, and potentially shallower soils.

## Statistical analysis for each climate zone

Data were pooled across years and a separate analysis was performed for each climate zone. We used conditional inference tree analysis to identify the most important factors explaining variation in farmer yields in each climate zone. This statistical method combines the ability to detect sources of variation and interactions in unstructured databases without losing interpretability[60]. Previous studies have used similar methods to identify yield constraints in farmer fields due to their appealing features for survey data analysis[10,67]. Tree-based methods do not have assumptions relative to data distribution, can handle missing data well, perform automatic variable selection (including interactions), and are robust against outliers[68]. Moreover, conditional inference trees outperform other tree-based methods regarding statistical inference because of their data partitioning strategy. Briefly, the algorithm only performs a data partition when there is a significant association between the response variable (i.e., yield) and the candidate explanatory variables (i.e., management practices and environmental covariables), selecting the variable with the strongest statistical association while considering the distributional properties of the variables[69]. In those climate zones with more than 150 observations (22 out of 25), we used a significance level of 0.01 to avoid spurious relations given the large sample sizes. The algorithm was further tuned to avoid overfitting by limiting the trees' depth and defining the minimum number of observations in intermediates and terminal nodes. We set a maximum tree depth of 10 nodes and ensured that each intermediate node contained more than 20% of the observations (or 200 when initial $n > 1000$) and 5% (or 50) for terminal nodes. We relaxed these parameters in Uganda and Nigeria due to their lower number of observations by setting a significance level of 0.1 and a minimum number of 10 and 8 observations for intermediate and terminal nodes, respectively. We summarized the findings from our regression trees analysis by calculating the frequency that a given management practice appears to have a significant effect on yield across the selected climate zone.

Non-manageable variables (elevation, soil type, topography, and precipitation) were included in the analysis but are not shown in Fig. 3 because the goal was to identify management practices that farmers can manipulate to increase their yields. Differences in agronomic management explained variation in maize yields in 95% of the climate zones, whereas variation in inter-annual precipitation, elevation, soil properties, and topography explained yield variation within climate zones in 64%, 32%, 24%, and 12% of the climate zones, respectively (Fig. S11a). These results are consistent with those obtained with the spatial machine-learning model (Fig. S11b). The fact that soil variables did not appear too frequently in our regression trees performed at climate zone level can be attributed to (i) relatively small variation in most soil properties within climate zones (Fig. S10), (ii) larger impact of agronomic practices and in-season weather on yield compared with soil properties (Fig. S12), and/or (iii) correlations between soil, terrain, and climate factors (Fig. S6). Another possibility is that current soil databases are not sufficiently granular and accurate enough for the purpose of field-level studies. Regardless of the exact reason, it is not likely to influence the main findings from our study, as our objective is not to determine whether climate, soil, or topography are more important sources for variation in maize yield within climate zones, but rather to identify the key agronomic management practices influencing yield after accounting for differences in biophysical background.

Likewise, we note that our statistical approach cannot fully separate the individual effects of highly correlated management variables[70], which is the case for N and P rates ($r = 0.72$), but not for any other pair of management practices (Figs. S6–S9). Therefore, the effect of higher N fertilizer rates could be interpreted as a combination of N and P, *i.e.*, better overall plant nutrition, rather than an effect of N alone.

## Assessment of yield gains due to improved crop management

Linear mixed-effect models were used to estimate the yield achieved by groups of farmers adopting different technological levels while accounting for the underlying environmental variation. We grouped farmers based on key management practices that strongly affected maize yields, as identified with the conditional inference trees. For key continuous variables, we categorized farmers based on which tercile of the variable distribution they belong to and selected those groups of farmers in either the lowest or highest tercile of each variable, defining their technological level based on their management practices combination. We compared the yield of (i) farmers that used open pollination varieties seed, were in the lowest tercile for N rate and plant density, and sown their fields on the average date or later regarding all other farmers in their same climate zone (baseline management), (ii) farmers incorporating hybrid seed and high N rate (highest tercile of N rate) but still using low densities and average to late sowing dates, (iii) hybrid seed, high N rate, and high plant density (highest plant density tercile) but average to late sowing dates, and (iv) hybrid seed, high N rate, high plant density, and early sowing dates (earliest sowing date deviation tercile).

We fitted a linear mixed-effect model with technological level category as fixed effect and climate zone and year as random effects (Eq. 1). We used a square root transformation of yield as the response variable to get normally distributed model residuals. Finally, we estimated the marginal yield mean for each technological level using the linear mixed-effect model to get each group's average yield over all possible environmental combinations in our database:

$$\sqrt{\text{yield}_i} = \beta_0 + \beta_{H,N} \bullet x_{H,N_i} + \beta_{H,N,P} \bullet x_{H,N,P_i} \\ + \beta_{H,N,P,S} \bullet x_{H,N,P,S_i} + u_{CZ_{j[i]}} + u_{\text{year}_{k[i]}} + \varepsilon_i \quad (1)$$

where $\sqrt{\text{yield}_i}$ is the square root of yield for maize field $i$; $\beta_0$ is the square root of yield for the baseline management (the intercept); $\beta_{H,N}$, $\beta_{H,N,P}$, and $\beta_{H,N,P,S}$ are the square root of yield improvement compared to the baseline for the technology levels incorporating hybrids ($H$), N fertilizer ($N$), higher plant densities ($P$) and early sowing dates ($S$); $x_{H,N_i}$, $x_{H,N,P_i}$, $x_{H,N,P,S_i}$ are dummy variables indicating to which technological level observation $i$ belongs, $u_{CZ_{j[i]}}$ is the random intercept for climate zone $j$; $u_{\text{year}_{k[i]}}$ is the random intercept of year $k$, and $\varepsilon_i$ is the error component for field $i$.

Finally, we estimated the monetary cost of adopting the highest technology level as the sum of their cost to farmers, based on One Acre Fund average seller prices in 2023 across four countries (Kenya, Rwanda, Tanzania, and Burundi), and One Acre Fund's operational costs per farmer, which is the cost of delivering these inputs to farmers, including training and credit services. We included the latter cost because One Acre Fund program operates at a loss, which is effectively subsidized by their donors. In other words, the total cost we computed includes what farmers currently pay plus the indirect monetary benefit they receive from One Acre Fund donors.

## Scenario assessment for maize production and self-sufficiency in Sub-Saharan Africa

We calculated the yield potential for rainfed maize in our study region and the whole SSA using data from the Global Yield Gap Atlas (GYGA, www.yieldgap.org). GYGA provides climate zone-specific actual and yield potential estimates for the main maize-producing regions in SSA.

When GYGA yield potential data were unavailable for a particular climate zone, we used the yield potential for the same climate zone in a neighboring country or country-level GYGA data. Additionally, to understand to which degree well-managed crops can approach the yield potential, we compared the GYGA-derived yield potential against the measured yield obtained in One Acre Fund maize variety trials in Rwanda. We selected maize plots planted early, with high plant density (>5 plants m$^{-2}$) and high N fertilizer rate (110 kg of inorganic N per hectare). Plots planted with open pollination varieties or non-commercial cultivars were discarded. Our final field trials dataset included 148 observations from three research stations (Kayonza, Karongi, and Rutsiro), three years (2020, 2021, and 2022), and 30 different cultivars. These plots received 15 tons per hectare of manure and 20 kg per hectare of inorganic P fertilizer. Maize yield was measured by sampling 6 m$^2$ in the center of each plot and reported at standard grain moisture content (15%).

We assessed the effect of the widespread adoption of improved agronomic practices on future SSA maize self-sufficiency, which is the ratio between maize production and demand in the region[4]. To do so, we estimated SSA maize production and demand for the year 2020 and for two scenarios by the year 2050: (i) the same rate of yield gain as in the last 20 years, and (ii) acceleration of yield gain rates to reach the same level of yield gap closure as the one achieved by farmers adopting improved management practices in our dataset. For current (2020) production and demand, we used FAOSTAT data between the years 2018 and 2020[1]. Current SSA maize demand was estimated based on maize annual production, imports, exports, and stock change. Maize demand in SSA by the year 2050 was estimated based on projected per-capita maize demand and population by 2050. In turn, per-capita maize demand was calculated based on current per-capita maize demand and the relative change in this parameter between 2020 and 2050 predicted by the International Model for Policy Analysis of Agricultural Commodities and Trade (IMPACT)[2]. Finally, the 2050 population was derived from the medium fertility variant (https://population.un.org/wpp/).

Because the goal was to assess the capacity of SSA to meet its demand for maize in existing areas under cultivation, we assumed the maize area to remain at 40 million ha between now and 2050 for our scenario assessment. The maize yield by 2050 under the same rate of yield gain scenario was calculated by assuming that the historical rate of yield gain rate (27 kg ha$^{-1}$ year$^{-1}$), calculated during the 2000–2020 period, will persist over the next 30 years. For the scenario of yield improvement, maize yield by 2050 was computed based on average rainfed yield potential in SSA derived from GYGA and the yield gap closure achieved by farmers adopting the highest technological level. The latter was calculated by comparing the yield of high-technology farmers (*i.e.*, those using hybrids, high N rate, and plant density, and sowing earlier) against the average yield potential in the 25 climate zones in our analysis. We note that the estimate from GYGA is based on all major climate zones where maize is produced in SSA, weighted according to their associated maize harvested area. Detailed calculations and results for the scenario assessment are provided in Supplementary Note 2.1. Considering that extrapolation of historical rates of yield gains over the next 30 years will not allow SSA to reach maize self-sufficiency on existing cropland by 2050, it is likely that larger maize areas and/or imports will be needed. Because of the large number of possible scenarios of cropland expansion and imports, we opt to simply estimate the amount of area or imports that would be needed in SSA to meet maize demand by 2050.

## Validation of yield analysis and extrapolation using machine learning

To validate the findings from our regression tree analysis, we used Gradient Boosting Machines (GBMs) to estimate the effect of individual agronomic management practices on maize yields across fields

and extrapolate the yield improvement associated with the potential adoption of these practices in the entire maize producing are in SSA. The GBMs is a machine-learning algorithm that develops an ensemble of shallow and weak sequential trees, in which each tree iteratively learns from the previous one[71]. We chose this algorithm rather than another one more frequently used with agronomic data, such as Random Forest[72], because of its ability to handle missing data. We trained the GBM model with all the field data in our farmer database and associated crop management and environmental variables as predictors (Tables S1 and S2). Since one of the objectives of the GBM model was to extrapolate our findings to other maize areas, we also included the following seasonal climatic variables in our model: average seasonal precipitation, growing-degree days, aridity index, and average temperature. We defined the maize growing season across SSA by compiling calendar information from GYGA and Crop Monitor (www.cropmonitor.org). To avoid overfitting the GBM model to our data regions, we tuned the model by selecting the parameters that resulted in the lowest spatial cross-validation errors by splitting the data into training and testing subsets with the k-fold nearest neighbor distance matching algorithm[73].

We estimated the effect of individual agronomic management practices on maize yields with the Shapley additive explanation (SHAP) method[74]. The SHAP values are the predicted impact of explanatory variables (agronomic practices, climatic conditions, and soil properties) on maize yields for each individual field, according to the GBM model. Finally, we predicted the maize yield of two extreme technology levels across maize areas with similar biophysical conditions to the one explored in our data. We defined the baseline management as late sowing (>10 days deviation from the climate zone-season average), low plant densities (<2 plants m$^{-2}$) and use of open pollination varieties, receiving no N fertilizer application, and pest control. Conversely, the intensified management included the use of hybrids, in-hole application of >60 kg ha$^{-1}$ of N, compost use, weeding, pesticide application, high plant density (5 plants m$^{-2}$), and early sowing date (15 days earlier than climate zone average). We constrained predictions to the rainfed maize area in SSA[75] with a dissimilarity index lower than 40%, which we estimated by considering the distance in environmental space to the most similar field in the database while weighting environmental variables according to their relative importance in the GBM model[76]. We note that models using the pooled data (i.e., without stratifying fields by climate zone) are subject to confounding effects because of the inherent correlation between some management practices and the biophysical background (e.g., higher plant density or N rate in favorable environments).

### Software
All data analysis was conducted in R (version 4.2.1) with the following packages and versions: terra (1.6.17), data.table (1.14.2), stringr (1.4.0), car (3.1.0), partykit (1.2.16), ltm (1.2.0), emmeans (1.8.5), lme4 (1.1.30), xgboost (1.7.5.1), treeshap (0.3.0), and CAST (0.8.1).

### Reporting summary
Further information on research design is available in the Nature Portfolio Reporting Summary linked to this article.

## Data availability
The field-level yield, management, and biophysical data generated in this study have been deposited in Zenodo (https://zenodo.org/doi/10.5281/zenodo.11115815)[77]. Data on yield potential from Global Yield Gap Atlas are available at www.yieldgap.org. Data on national average maize yield, harvested area, production, export, and import, and demand from FAOSTAT are available at www.fao.org/faostat. Data on maize distribution from SPAM map are available in www.mapspam.info. Data on population size from the UN are available at https://population.un.org/wpp/. Data on per-capita future maize

demand is available at https://www.ifpri.org/project/ifpri-impact-model. Spatial vector data with administrative borders is available at https://gadm.org/. Source data are provided with this paper.

## Code availability
The R code for the current study is publicly on GitHub: https://github.com/AramburuMerlos/SSA_maize_management

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

## Acknowledgements

We are thankful to Ken Giller (Wageningen University), Charles Wortmann (University of Nebraska-Lincoln), and Rob Moss (Independent Consultant) for useful comments on an earlier draft of this manuscript. This project was supported by the Bill and Melinda Gates Foundation through the Niche project (INV-030103). We are also grateful to the One Acre Fund personnel and farmers who participated in the survey.

## Author contributions

F.A.M. and P.G. conceived and designed the study. N.M., A.S., and S.A. provided and compiled the data analyzed in this study. F.A.M.T. and J.J.O. run quality control measures on the data and estimate auxiliary variables for the analysis. F.A.M. performed the spatial analysis and data analysis. F.A.M. and P.G. wrote the paper, with contributions from other authors.

## Competing interests

The authors declare no competing interests.
