## [Peer Review File · Nature Communications]

Reviewers' Comments:

Reviewer #1:

Remarks to the Author:

This study investigated the possibility of achieving food self-support in SSA without massive cropland expansion. Drawing from an extensive dataset of 13,364 smallholder fields, the study underscores the potential of optimal agronomic practices. By emphasizing nutrient management, crop management, and cultivar selection, the research suggests that yields can more than double, producing an additional 93 million tons within existing croplands. The manuscript strongly advocates for a reorientation in agricultural research and development towards these high-impact practices.

While I find the topic important and the large dataset certainly valuable, I think the analysis presented is quite descriptive, and figures like Fig.1 and Fig.2 do not really bring in "new and important results". Fig 3 and 4 are slightly better, but they don't seem to be at the same level as those I would expect to see in Nature Communications. Given the data, I think the authors have the chance to present a much more comprehensive and quantitative analysis.

Promoting good agronomic practices is not something new in SSA, and there are many existing programs reporting mixed results regarding the magnitude of benefits, which tend to be small and lead to the termination of programs. So what's interesting is why this One Acre Fund data showed very large impacts. What has the program done right or is it generally applicable to the vast remaining SSA regions? These facts should be brought up front and analyzed more.

Overall, I feel the manuscript in its current shape is obviously below the standard of Nature Communications, but the dataset has good potential. So I would like to see a resubmission with substantial revision.

Some more specific comments:

L23: results like "more than double" should be reported with caution and reflect uncertainty.

L40: a quick thought on "imports", how about optimizing maize to where they can grow favorably (higher yield, higher WUE, and NUE) and easier to access inputs? This can be argued easily but needs more background information in the introduction.

L29: Improving agronomy practices is not a new topic for Africa and there are many efforts/publications to reference. However, most of these programs found small treatment effects, which unfortunately led to the backout of many programs. Also, logistic constraints are major barriers to scaling up. This existing knowledge should be discussed more in the introduction to better position this niche of this study.

L80: To help interpret the results, the authors need to provide more details at least the resolution of these data. Especially for climate data, the relatively low resolution may not support a robust clustering analysis. Here, I am talking about the real information resolution, not the nominal resolution of interpolated products.

L81: I can't agree with "state of art statistical methods". Seems just ordinary methods were implemented.

L84-85: This is quite confusing. Fig S2 is about yield spread, good to know but similar distribution is not unexpected. Fig S1 (which is hard to interpret visually) shows all sites are in homologous climate zones. If you say other climates are similar, then why do the clustering? if you admit variation and use clustering, then this homologous CZ map basically tells us nothing, and instead, you would rather show GDD and aridity index maps or distributions.

L91: how robust is the climate zone clustering and how it's used in subsequent analysis is unclear, making it look like a stand-alone analysis.

L93: this type of technical jargon should be avoided for the broader audience.

L99-101: I didn't find specific data supporting this argument.

L101-102: similarly, where's the data for this statement?

L109: I'm sure most readers will be confused by "variable appearance in trees". Please rephrase with some plain text.

L111: what's the definition/threshold for "explained"?

L114: how's "the importance" estimated? better to give some very concise explanations to help interpret results without going through the methods.

L138: the up/down arrows often mean increase/decrease, and don't represent the high/low of a variable. It's a bit misleading now. Simply add "high", or "low" since you have enough space here.

L160: the results of linear mixed based on experiment data should be shown as one main result to increase transparency and credibility.

L161: better to reference earlier in the main text as well.

L177: is there 27.3 Mha of land available in SSA without invading protected areas?

L213-215: unclear from the manuscript how this is achieved

L222-224: what's the cost of doing so and are they logistically feasible for now, and if not, how to implement?

L241-242: please see my comments early in the introduction, and should be also addressed along with the extrapolation estimation

L248-249: I feel this message is already clear across SSA and most countries already have a large training system to promote good practices, but the tricky part is: 1) many ICT trials or control experiments didn't show a significant difference or up to 50% at most, I barely see results of doubling yield; 2) farmers don't always follow the guidance, and data is super noisy; 3) there lacks cost-effective, accurate approach to track progress at scale.

L315: move this sentence to the beginning of the paragraph

L327-328: what's the process to group CZs and how robust is the approach given that the weather inputs are essentially interpolated products even though the nominal resolution is high? More details are needed

L341-346: how these data were used to "account for" residual spatial and temporal variation was not clearly explained. More quantitative descriptions can help

L350: better to mention this early in the main text as well

L384-385: does that mean you used a vector of the tercile of each variable to describe a farmer?

L391-392: Please describe your linear mixed model with explicit equation form and clarify your variables

L404: I don't understand this sentence, please elaborate more

L434: but how about climate change reducing yield?

Reviewer #2:

Remarks to the Author:

The paper uses a large dataset containing on-farm measured maize yields to reveal which agronomic practices are most efficient to increase yields across a range of biophysical conditions. Based on the results of the statistical analysis, a scenario analysis is performed to explore the prospects of meeting future maize demands in sub-Saharan Africa.

The article claims to cover maize-growing areas of the entire sub-Saharan African (SSA) subcontinent. In reality the scope is more limited and based on a database covering three regions across 4 countries located in East Africa. Because the extrapolation to SSA is both not convincing and insufficiently documented, I'm afraid I cannot accept the paper.

The extrapolation of the results from the East African countries to the entire maize-growing area in SSA is not convincing and the methods used are insufficiently documented.

Firstly, it is mentioned that the extrapolation is done based on climate and soil information, but it seems that for soil information only the plant available water holding capacity and the topography wetness index are taken into account. The crop response to agronomic inputs and practices depends on a wider range of soil-related characteristics (e.g. soil organic carbon content, nutrient contents, pH, texture) (see e.g. Rurinda et al., 2020, Kihara et al., 2016), which seem to be omitted in this case. Hence the extrapolation based on a limited set of environments covering only East Africa to the entire region of SSA is not sound.

Secondly, the methods used are insufficiently documented. For example, it is unclear how the years are treated in the conditional inference tree analysis and in the regression analysis. The same applies to the farmers: since multiple years are included in the analysis, it is likely that information from the same farmers, and even fields, is included in the database. How is the farmer/field factor treated in the statistical analysis? With respect to the linear mixed effect models, it is said that "underlying environmental variation is accounted for" (line 380-381) and that "you control for confounding factors related to climate, soil and terrain" (line 214). But how exactly is this done? The method section contains insufficient information for the study to be repeatable. For example, the authors could give the model equation that is used to improve transparency.

Thirdly, also the description of the method used for the scenario analysis leaves many questions. For example, it seems that a different method is used for estimating current and future demand. Is that indeed the case, and why? What is the implication of using a different method for the interpretation of the results?

Finally, the authors claim that yield estimates are more reliable than farmer-reported yields, because they are measured. While this is indeed the major strength of this paper, it leaves one wondering about the quality of the farmer-reported other variables related to agronomic management (line 289). While this is hard to further improve, at least more information on the method of data collection needs to be given and the authors should reflect on the implications of uncertainty in these variables. Also, with respect to field size, which is notoriously difficult to estimate based on farmer reports, it is said that this was verified by One Acre Fund staff (line 294). How was it verified, and if it was measured, why not use the measured field size?

Detailed comments:

- Lines 50-55: to increase the rate of yield gain in maize, we don't need new approaches to identify agronomic practices that can improve on-farm yield. The agronomic practices are known, but the biggest problem is the adoption rate by farmers. Please rephrase this argument and articulate more clearly why new approaches are needed, and in what way existing approaches fall short in achieving this.
- Table 1, the SSR for the current situation is about 1. However, as this estimate includes import and export, can you really speak about self-sufficiency?
- Lines 188: you mention the strong focus on practices by the range of approaches. This is also what you do in your study (focus on practices, in this case agronomic practices), so this is confusing. Consider rephrasing this statement.
- Line 243: you refer to van Loon et al. (2019) to back up your findings, but please note that in this study in Ghana, there was no consistency across methods in the factors that influenced on-

farm yields.

References:

- Kihara, J., Huising, J., Nziguheba, G., Waswa, B.S., Njoroge, S., Kabambe, V., Iwuafor, E., Kibunja, C., Esilaba, A.O., Coulibaly, A., 2016. Maize response to macronutrients and potential for profitability in sub-Saharan Africa. *Nutr. Cycl. Agroecosystems* 105, 171–181. <https://doi.org/10.1007/s10705-015-9717-2>
- Rurinda, J., Zingore, S., Jibrin, J.M., Balemi, T., Masuki, K., Andersson, J.A., Pampolino, M.F., Mohammed, I., Mutegi, J., Kamara, A.Y., Vanlauwe, B., Craufurd, P.Q., 2020. Science-based decision support for formulating crop fertilizer recommendations in sub-Saharan Africa. *Agric. Syst.* 180, 102790 <https://doi.org/10.1016/j.agsy.2020.102790>.

Responses to Reviewers' Comments (and our responses in red)

We thank the editor and the two reviewers for their comments on our paper. We have constructively addressed all their comments in the revised MS. Our point-by-point responses to the comments are shown below in red. Briefly, major changes include (i) addition of other maize producing areas in Nigeria, Zambia and Uganda into our analysis, (ii) addition of new analysis based on machine learning methods to cross validate the results from our analysis based on regression trees, (iii) moving descriptive figures (Fig 1-2 in the original MS) as extended figures and addition of a new figure summarizing the results from the machine learning methods, (iv) improving the extrapolation of the results from our study areas to entire maize producing area in SSA, and (v) further elaborating on the methodology used for clustering fields and inclusion of more variables into our statistical analysis.

Comments from Reviewer #1:

This study investigated the possibility of achieving food self-support in SSA without massive cropland expansion. Drawing from an extensive dataset of 13,364 smallholder fields, the study underscores the potential of optimal agronomic practices. By emphasizing nutrient management, crop management, and cultivar selection, the research suggests that yields can more than double, producing an additional 93 million tons within existing croplands. The manuscript strongly advocates for a reorientation in agricultural research and development towards these high-impact practices.

While I find the topic important and the large dataset certainly valuable, I think the analysis presented is quite descriptive, and figures like Fig.1 and Fig.2 do not really bring in "new and important results". Fig 3 and 4 are slightly better, but they don't seem to be at the same level as those I would expect to see in Nature Communications. Given the data, I think the authors have the chance to present a much more comprehensive and quantitative analysis.

Response: We appreciate that the reviewer recognizes the importance of the topic. Following reviewer's suggestion, we moved Figures 1-2 as Extended Data Figure and added a new figure summarizing the findings from an additional analysis using machine learning models (Figure 2 in the revised MS). We still kept the original Figures 3-4 (now Figures 1 & 3 in the revised MS) in the main text as they will help non-experts to understand the finding from the statistical analyses.

Promoting good agronomic practices is not something new in SSA, and there are many existing programs reporting mixed results regarding the magnitude of benefits, which tend to be small and lead to the termination of programs. So what's interesting is why this One Acre Fund data showed very large impacts. What has the program done right or is it generally applicable to the vast remaining SSA regions? These facts should be brought up front and analyzed more.

Response: This suggestion is well taken. However, it was not our goal to analyze the success (or not) of a specific program (in this case, One Acre Fund). That would require a yield comparison of what the One Acre Fund promotes versus a reference farmer management--- this was not the objective of our study. Thus, whereas the suggestion made by the reviewer #1 sounds like an interesting idea to be explored on a separate paper, we found it difficult to consider in the context of this study. Instead, our goal was to demonstrate how a data-driven approach, based on the data

collected *via* the One-Acre Fund program (10,000+ field observations across 25 production environments), can help to identify combinations of management practices that lead to consistent increases in on-farm yield for each agro-ecological zone. In turn, at a time in which essential things sometimes get invisible to the eyes, we believe it is vital to show and/or remind donors, charitable foundations, and international and national research organizations and programs, that the agronomic means to increase yields in SSA already exist (and can be identified for a given production environment as we have done) and the challenge is therefore how to promote these technologies and getting them adopted farmers instead of trying to find ‘new’ alternative approaches that lack empirical validation such as agro-ecology, regenerative agriculture, and others, as we mentioned in the discussion of our study (L 209-231 of revised MS).

Overall, I feel the manuscript in its current shape is obviously below the standard of Nature Communications, but the dataset has good potential. So I would like to see a resubmission with substantial revision.

Some more specific comments:

L23: results like "more than double" should be reported with caution and reflect uncertainty.

Response: Followig the reviewer’s suggestion, we rephrased the sentence (L 25 of revised MS)

L40: a quick thought on "imports", how about optimizing maize to where they can grow favorably (higher yield, higher WUE, and NUE) and easier to access inputs? This can be argued easily but needs more background information in the introduction.

Response: We agree with the reviewer comment. Based on this comment, we have added a comment in the discussion section (L 259-260 of revised MS).

L29: Improving agronomy practices is not a new topic for Africa and there are many efforts/publications to reference. However, most of these programs found small treatment effects, which unfortunately led to the backout of many programs. Also, logistic constraints are major barriers to scaling up. This existing knowledge should be discussed more in the introduction to better position this niche of this study.

Response: Thanks for this comment. Following the suggestion, we have added text to the Introduction to better position our study (L 54-56 of revised MS). Note that we also dicuss our findings in the context of exisiting knowledge in the discussion section (L 230-249 of revised MS).

L80: To help interpret the results, the authors need to provide more details at least the resolution of these data. Especially for climate data, the relatively low resolution may not support a robust clustering analysis. Here, I am talking about the real information resolution, not the nominal resolution of interpolated products.

Response: As requested, the spatial resolution of each data source has been added to and in Methods (Supplementary Table S2). In all cases, we made our best effort to use the best available data for SSA (admittedly still far from perfect). For example, we redlined the CZ using higher resolution data (L. 395-396 of revised MS) and added high-resolution elevation data as a covariable to account for more granular differences (L. 408-410 of revised MS). We believe we have done a decent job in selecting and using spatial biophysical data.

L81: I can't agree with "state of art statistical methods". Seems just ordinary methods were implemented.

Response: We rephrased the sentence as requested (L. 77-80 of revised MS). Note that our analysis now includes machine learning techniques, which can be considered "state of art".

L84-85: This is quite confusing. Fig S2 is about yield spread, good to know but similar distribution is not unexpected. Fig S1 (which is hard to interpret visually) shows all sites are in homologous climate zones. If you say other climates are similar, then why do the clustering? if you admit variation and use clustering, then this homologous CZ map basically tells us nothing, and instead, you would rather show GDD and aridity index maps or distributions.

Response: Based on this comment, we now show (i) comparison of seasonal weather variables and soil properties between our field database and those across the SSA maize area growing area using density plots (new Supplementary Fig. S1) and (ii) dissimilarity in biophysical variables between SSA maize-producing areas and the maize fields in our database (new Supplementary Fig. S2).

L91: how robust is the climate zone clustering and how it's used in subsequent analysis is unclear, making it look like a stand-alone analysis.

Response: based on this comment, we have added text to Methods explaining how we used the CZs for field clustering and justifying the approach (L 389-402 of revised MS)

L93: this type of technical jargon should be avoided for the broader audience.

Response: we excluded the technical jargon as suggested (L 109-111 of revised MS)

L99-101: I didn't find specific data supporting this argument.

Response: we rephrased the sentence and added a reference to support it (L 98 of revised MS)

L101-102: similarly, where's the data for this statement?

Response: the text was rephrased in response to other comment and, thus, there is no need to address this comment as the text in question was excluded from the revised MS.

L109: I'm sure most readers will be confused by "variable appearance in trees". Please rephrase with some plain text.

Response: we rephrased the sentence as requested (Fig. 1 in revised MS)

L111: what's the definition/threshold for "explained"?

Response: as requested, we rephrased the text for clarity (L 104-105 of revised MS)

L114: how's "the importance" estimated? better to give some very concise explanations to help interpret results without going through the methods.

Response: as requested, we rephrased the text for clarity (L 109-111 of revised MS)

L138: the up/down arrows often mean increase/decrease, and don't represent the high/low of a variable. It's a bit misleading now. Simply add "high", or "low" since you have enough space here.

Response: we modified the figure as requested (Fig 3 of revised MS).

L160: the results of linear mixed based on experiment data should be shown as one main result to increase transparency and credibility.

Response: The results from the linear mixed effect model were already shown in Figure 4 of the original MS. Thus, no change was made in the revised MS in response to this comment.

L161: better to reference earlier in the main text as well.

Response: as requested, we added an explanation on how the extrapolation of our results to the rest of SSA was performed and also explained how we cross-validated the results using a machine-learning method (L 175-183 of revised MS)

L177: is there 27.3 Mha of land available in SSA without invading protected areas?

Response: The quick answer is that the most productive land is already under production, thus, a substantial fraction of new cropland expansion will likely occur at expense of natural ecosystems and/or marginal environmental for crop production. Based on this comment, we elaborate on this in the MS (L 190-192 & of revised MS).

L213-215: unclear from the manuscript how this is achieved.

Response: see our previous responses. We believe that it is now clear how the clustering was done.

L222-224: what's the cost of doing so and are they logistically feasible for now, and if not, how to implement?

Response: this is well taken, but we prefer to leave the sentence open as it is now because, while we recognize the importance of focussing on better agronomy, the specific means to do it is context-specific and we prefer to avoid the temptation of being too prescriptive. No change was made in the revised MS in responses to this comment.

L241-242: please see my comments early in the introduction, and should be also addressed along with the extrapolation estimation.

Response: see our previous responses on this.

L248-249: I feel this message is already clear across SSA and most countries already have a large training system to promote good practices, but the tricky part is: 1) many ICT trials or control experiments didn't show a significant difference or up to 50% at most, I barely see results of doubling yield; 2) farmers don't always follow the guidance, and data is super noisy; 3) there lacks cost-effective, accurate approach to track progress at scale.

Response: see our previous response to a similar comment. We rephrased the text in many parts of the discussion section in response to this comment (L 214-217 & L 289-290 of revised MS), also discussing the limitations from previous studies (L 230-249 of revised MS) and highlighting the challenges for adoption (L. 302-310). We appreciate the comments from Reviewer #1 as they helped to put together a more balanced discussion section.

L315: move this sentence to the beginning of the paragraph.

Response: We moved the sentence as requested (L 361-362 of revised MS).

L327-328: what's the process to group CZs and how robust is the approach given that the weather inputs are essentially interpolated products even though the nominal resolution is high? More details are needed.

Response: As requested, we now provide more details on how we grouped CZs and the approach used for the weather data (L 389-403 of revised MS)

L341-346: how these data were used to "account for" residual spatial and temporal variation was not clearly explained. More quantitative descriptions can help.

Response: As requested, we added more details on these methods (L 406-411 of revised MS)

L350: better to mention this early in the main text as well.

Response: As requested, we now mentioned that we used conditional inference tree analysis in the main text as well (L 89 and 111 of revised MS)

L384-385: does that mean you used a vector of the tercile of each variable to describe a farmer?

Response: In our approach, we filtered fields based on agronomic practices to keep those fields that fall in each category combination. Based on this comment, we added text to explain better our approach and avoid confusion (L 466-469 of revised MS).

L391-392: Please describe your linear mixed model with explicit equation form and clarify your variables.

Response: We added the equation as requested (L 489-492 of revised MS)

L404: I don't understand this sentence, please elaborate more.

Response: We deleted the sentence in question from the revised MS.

L434: but how about climate change reducing yield?

Response: As requested, we elaborated on the impact of climate change (L 295-300 of revised MS)

Comments from Reviewer #2:

The paper uses a large dataset containing on-farm measured maize yields to reveal which agronomic practices are most efficient to increase yields across a range of biophysical conditions. Based on the results of the statistical analysis, a scenario analysis is performed to explore the prospects of meeting future maize demands in sub-Saharan Africa.

The article claims to cover maize-growing areas of the entire sub-Saharan African (SSA) subcontinent. In reality the scope is more limited and based on a database covering three regions across 4 countries located in East Africa. Because the extrapolation to SSA is both not convincing

and insufficiently documented, I'm afraid I cannot accept the paper.

Response: we have constructively addressed this comment by (i) explaining better the original extrapolation method that we used and adding a cross-validation of the approach using a machine-learning method (see L 175-183, last section of the Methods & Section S1 of supplementary material), (ii) adding new data from other countries (Zambia, Nigeria, and Uganda) to the analysis (see Extended Data Figure 1), and (iii) adding a more detailed comparison of the environmental variables in our study areas in relation to those across the entire maize growing area in SSA (Supplementary Fig S1). We thank the Reviewer #2 for their help to improve our methodology description and justify better why we believe our extrapolation is valid. See responses below to the more detailed comments on this issue.

The extrapolation of the results from the East African countries to the entire maize-growing area in SSA is not convincing and the methods used are insufficiently documented.

Firstly, it is mentioned that the extrapolation is done based on climate and soil information, but it seems that for soil information only the plant available water holding capacity and the topography wetness index are taken into account. The crop response to agronomic inputs and practices depends on a wider range of soil-related characteristics (e.g. soil organic carbon content, nutrient contents, pH, texture) (see e.g. Rurinda et al., 2020, Kihara et al., 2016), which seem to be omitted in this case. Hence the extrapolation based on a limited set of environments covering only East Africa to the entire region of SSA is not sound.

Response: Thanks for this comment as it gave us an opportunity to extend the analysis to other production environments, which together with a more detailed description of the ranges of environmental factors (and inclusion of new ones) helped us to justify better our extrapolation methods and, ultimately, improve the quality of our MS.

First, we extended our analysis to other countries, including Nigeria, Zambia, and Uganda, to increase our environmental range (see new Extended Figure 1 of the revised MS). Results were consistent with those from the climate zones included in the original MS (see regression trees #24 and #25 in Supplementary Fig. S8). We also cross validated the results from the regression tree analysis with results from a new analysis based on machine-learning methods and we show that results are consistent between the two approaches (see new Figure 2 of the revised MS).

Second, as requested, we included other soil properties in our analysis as covariables, such as soil organic carbon, pH, CEC, and clay content, and we showed that the range of soil properties explored by our database was similar to that across the whole maize producing area in SSA (Supplementary Fig. S1). Inclusion of the new variables into our regression tree analysis (and new machine learning analysis) did not modify the main findings from our original MS and did not improve substantially the explanatory power of our models.

Third, we used a machine learning model to extrapolate the results from our study areas to the rest of the maize producing areas in SSA based on climate and soil variables and additional information on crop calendars from the other cropping systems in SSA. Extrapolation through the machine learning model gave nearly identical results to the ones described in the original MS based on direct extrapolation (L 175-186 of revised MS and referenced cited therein). Thus, consistency in the results from both extrapolation methods, and inclusion of new maize producing areas (e.g.

Nigeria), gave confidence that our findings are relevant for the whole SSA and not just for our study areas.

Secondly, the methods used are insufficiently documented. For example, it is unclear how the years are treated in the conditional inference tree analysis and in the regression analysis. The same applies to the farmers: since multiple years are included in the analysis, it is likely that information from the same farmers, and even fields, is included in the database. How is the farmer/field factor treated in the statistical analysis? With respect to the linear mixed effect models, it is said that “underlying environmental variation is accounted for” (line 380-381)” and that “you control for confounding factors related to climate, soil and terrain” (line 214). But how exactly is this done? The method section contains insufficient information for the study to be repeatable. For example, the authors could give the model equation that is used to improve transparency.

Response: Thanks for these comments. We made several changes in response to them. First, we explained now in the methods section that year effects are accounted for by their seasonal precipitation in the conditional inference trees (L 409-411 of revised MS), while year was considered a random effect in the linear mixed effect model (L 150-152, L 161-162 & 476-477 of revised MS). Unfortunately, given the small size of the plots and geolocation information that were available, we are not able to determine whether a field was consistently the same over time or a different one (considering their small sizes). Hence we could not include ‘field’ as a factor – still, we do not consider it a major limitation given the large number of observations in our data. We added text on this in the discussion to elaborate about how this uncertainty might affect our findings (L 281-285 of revised MS). Additionally, we elaborated in the methods to make clear what variables were considered random and fixed variables in the linear mixed effect model, and we also provided the equation and more details in Methods (L 483-491 of revised MS). Likewise, we also added references to justify the stratification approach by CZ followed in our study (L 389-393 of revised MS). We want to thank reviewer #2 for helping us to be more detailed and transparent about our methods.

Thirdly, also the description of the method used for the scenario analysis leaves many questions. For example, it seems that a different method is used for estimating current and future demand. Is that indeed the case, and why? What is the implication of using a different method for the interpretation of the results?

Response: This is well taken. We do not see a problem in using different methods for estimating current and future demand. Indeed, many previous studies have used the same approach that we followed here. Briefly, in the case of current demand, it is relatively easy, as one needs to know the current production, the imports and exports, and the annual balance and these data are available through FAOSTAT. In the case of future demand, we used the relative change in demand per capita as reported by IMPACT model and we estimated the overall demand using the projected population data from United Nations. Thus, no change was made in the methodology as we believe it is robust and consistent with previous studies, but we rephrased the text in the methods for clarity (L 517-525 of revised MS).

Finally, the authors claim that yield estimates are more reliable than farmer-reported yields, because they are measured. While this is indeed the major strength of this paper, it leaves one wondering about the quality of the farmer-reported other variables related to agronomic

management (line 289). While this is hard to further improve, at least more information on the method of data collection needs to be given and the authors should reflect on the implications of uncertainty in these variables. Also, with respect to field size, which is notoriously difficult to estimate based on farmer reports, it is said that this was verified by One Acre Fund staff (line 294). How was it verified, and if it was measured, why not use the measured field size?

Response: In response to this comment, we elaborated in the discussion on the uncertainty in farmer-reported data and how it may (or not) affect our conclusions (L 281-285 of revised MS). We also elaborated on the protocols to collect field size data (L 349-352 of revised MS). Briefly, despite uncertainty on field size reported by farmers, the fact that the database included 13,000+ observations and that the results were consistent among a large number of production environments, give us confident that the results are robust despite the inherent uncertainty of the farmer-reported data.

Detailed comments:

- Lines 50-55: to increase the rate of yield gain in maize, we don't need new approaches to identify agronomic practices that can improve on-farm yield. The agronomic practices are known, but the biggest problem is the adoption rate by farmers. Please rephrase this argument and articulate more clearly why new approaches are needed, and in what way existing approaches fall short in achieving this.

Response: Following reviewer's suggestion, we rephrased the sentence (L 54-56 of revised MS)

- Table 1, the SSR for the current situation is about 1. However, as this estimate includes import and export, can you really speak about self-sufficiency?

Response: The SSR is the ratio between production and demand. Hence, a ratio of one means that countries are roughly producing what they produce. Based on this comment, we rephrased the caption of Table 1 for clarity (L 199-200 of revised MS).

- Lines 188: you mention the strong focus on practices by the range of approaches. This is also what you do in your study (focus on practices, in this case agronomic practices), so this is confusing. Consider rephrasing this statement.

Response: We rephrased our statement as requested (L 214-217 of revised MS)

- Line 243 refers to van Loon et al. (2019) to back up findings, but please note that in this study in Ghana, there was no consistency across methods in the factors that influenced on-farm yields.

Response: Based on this comment, we replaced this reference by more appropriate ones (L 280 of revised MS)

Reviewers' Comments:

Reviewer #1:

Remarks to the Author:

I have carefully reviewed the revised manuscript and feel the quality of this version is much improved. While appreciable efforts have been made since the previous submission, there are still some significant concerns that need to be addressed before I can recommend acceptance. These concerns are detailed below:

1. Misinterpretation of Comment on One Acre Fund Data:

It appears there has been a misinterpretation of my earlier comment regarding the data from One Acre Fund. My focus is not on the success metrics of 1AF, but rather on the robustness of the results. The authors have added some discussion and highlighted the primary distinction of their study being the scale of data utilized. However, a critical concern remains unaddressed: the extent to which the observed improvements in relationships (as depicted in the study) are attributable to larger spatial variations. Such variations could potentially overshadow the impacts of agronomic management practices. This aspect needs to be thoroughly investigated and clarified in the manuscript.

2. Standard Error Concerns in Figure 3

The standard error (SE) depicted in Figure 3 seems to be considerably smaller than what I have observed in a similar subset of 1AF data previously used. This discrepancy raises questions about the data analysis and interpretation methods employed. Additionally, I recommend that the authors refer to Burke & Lobell's 2017 PNAS publication for a comparative perspective on the effect of fertilizer and hybrid seeds, which seem to be much noisier. A reevaluation is necessary to ensure the robustness and credibility of the findings.

3. "State of art" methodology

The manuscript refers to the use of Gradient Boosting Machines (GBM) or Random Forest as "state of art" methodologies. While these methods are valuable, classifying them as the forefront of machine learning technology might be misleading. "state of art" often implies the use of the most advanced techniques available, which is not the case here. I am not suggesting a shift to more complex machine learning or deep learning algorithms, as the appropriateness of the method is paramount. However, the current portrayal in the manuscript might overstate the sophistication of the employed methods.

4. Causal Analysis

The analysis in the paper seems to predominantly focus on correlation and predictive performance rather than causal inference. Given that the primary objective is to understand the drivers rather than merely develop predictive models, a transition to causal inference is essential. In fact, GBM is capable of contributing to causal inference, but this aspect has not been sufficiently explored in the current manuscript. Incorporating and elucidating a causal analysis approach would significantly enhance the depth and impact of the research.

Reviewer #3:

Remarks to the Author:

Sub-Saharan Africa needs better agronomy to meet maize demand without massive cropland expansion and imports (NCOMMS-23-33461A-Z)

I have gone through the paper and the authors' response to the reviewers' comments, specifically to reviewer #2. I agree that the paper used valuable dataset collected from African smallholder farms and the analysis of such data would add value to inform the global community at large and the agricultural literature specifically.

In their response, the authors have tried to clarify and/or provide additional information and data/analysis to the issues raised by the reviewers and this has improved the quality and

relevance of the paper. However, there are outstanding issues that the authors didn't address fully, or their responses raised additional questions or concerns, particularly to the issues raised by reviewer #2.

There are two major concerns that need additional clarification from the authors.

1. In their response to reviewer #2's concern on missing soil parameters, the authors stated "... as requested, we included other soil properties in our analysis as covariables, such as soil organic carbon, pH, CEC, and clay content, and we showed that the range of soil properties explored by our database was similar to that across the whole maize producing area in SSA (Supplementary Fig. S1). Inclusion of the new variables into our regression tree analysis (and new machine learning analysis) did not modify the main findings from our original MS and did not improve substantially the explanatory power of our models".

This statement raises a major concern on the sensitivity of the model/methodology used for analysis. In a region with diverse terrain, soil type and soil nutrient conditions, it is hardly possible to believe that carbon, pH, CEC and soil texture parameters do not affect a given analysis across the region. To support this claim, the authors need to show evidence that their method/model is sensitive to input parameters.

2. There is no lack of evidence on the role of good agronomic practices in increasing productivity in SSA, and the findings from this paper are not new. The authors' response on the concern raised by reviewer #2 regarding the full adoption assumption of the identified agronomic practices to project future production levels in SSA is not convincing. Although the potential to increase maize productivity through increasing the level of inorganic fertilizers, use of hybrid seeds and proactive crop protection practices is clear to smallholders farmers and extension workers, the major challenge is that most smallholders do not have the required financial resources, credit services, agricultural insurance and market access to apply the inputs and hence the low adoption of improved seeds, higher fertilizer rates and use of agro-chemicals across the region. The lack of consideration of these adoption constraints led the authors to project a doubling of maize productivity in SSA, which is unrealistic. I think the authors could consider different adoption levels (which could vary among sub-regions) in their future production projection so that their estimates come closer to "reality". This also provide readers, particularly those in the region, to consider what is possible/feasible based on their knowledge and experience.

3. Since the authors claim a doubling of maize productivity in SSA with the use of agronomic practices they identified (which could motivate donors and governments), reader would want to know the cost of full or partial implementation of the recommendations. It would make the paper useful and informative, if the authors could give an estimation of the investment required to adopt the practices, and discuss it relative to on-going investments in the region and the gap that needed to be filled.

Responses to Reviewers' Comments (and our responses in red)

We thank the editor and the two reviewers for their comments on our paper and overall support for publication after revision. We have constructively addressed all their comments in the revised MS. Our point-by-point responses to the comments are shown below in red.

Comments from Reviewer #1:

I have carefully reviewed the revised manuscript and feel the quality of this version is much improved. While appreciable efforts have been made since the previous submission, there are still some significant concerns that need to be addressed before I can recommend acceptance.

Response: we appreciate the positive feedback from Reviewer #1 about the revised MS and support for publication after addressing remaining concerns. See our detailed responses below.

These concerns are detailed below:

1. Misinterpretation of Comment on One Acre Fund Data:

It appears there has been a misinterpretation of my earlier comment regarding the data from One Acre Fund. My focus is not on the success metrics of 1AF, but rather on the robustness of the results. The authors have added some discussion and highlighted the primary distinction of their study being the scale of data utilized. However, a critical concern remains unaddressed: the extent to which the observed improvements in relationships (as depicted in the study) are attributable to larger spatial variations. Such variations could potentially overshadow the impacts of agronomic management practices. This aspect needs to be thoroughly investigated and clarified in the manuscript.

Response: We appreciate that the reviewer thinks this version has much better quality. We apologize if we were not sufficiently clear in the original MS to explain how we accounted for variation in climate and soil. We stratified fields by agro-climatic zones to account for environmental gradients over larger region, which considerably narrowed the ranges in soil and climate variables within each of them, and subsequently performed a separate analysis to identify management factors with greatest influence on crop yield for each climate zone, which also included field-specific soil, terrain, and in-season weather variables to account for any residual variation in these variables. Based on this comment, we added text to explain better how we accounted for the environmental variation in our analyses (L 81-83 of revised MS).

2. Standard Error Concerns in Figure 3

The standard error (SE) depicted in Figure 3 seems to be considerably smaller than what I have observed in a similar subset of 1AF data previously used. This discrepancy raises questions about the data analysis and interpretation methods employed. Additionally, I recommend that the authors refer to Burke & Lobell's 2017 PNAS publication for a comparative perspective on the effect of fertilizer and hybrid seeds, which seem to be much noisier. A reevaluation is necessary to ensure the robustness and credibility of the findings.

Response: Standard errors (SEs) for the yellow bars in Figure 3 (average maize yields for groups of fields following similar management practices) were derived from the linear mixed-effect model depicted in Equation 1, which considered climate zone and year as random effects. We estimated them in R using the functions `lme4::lmer()` to fit the linear mixed-effect model and `emmeans::emmeans()` to compute the estimated marginal means and their respective SEs. Therefore, these SEs show the expected variation in average maize yields for farmers adopting the same technology within the same climate zone and year. The SE for the on-station trial yields (blue bar in Figure 3) was calculated based on data from One-Acre Fund maize variety trials. We considered 148 well-managed trials, that were planted early with commercial hybrids, proper plant stands (> 5 plants m⁻²), and receiving high fertilizer rates (110 kg N fertilizer per ha, 20 kg P fertilizer per ha, and 15 t/ha manure). The dataset included observations from three research stations, three years, and 30 different cultivars. In response to this comment, we added text in the caption to Figure 3 legend to explain how SEs were calculated (L 156-160 of revised MS).

We agree with Reviewer #1 that Burke & Lobell (PNAS, 2017) found a very noisy correlation between maize yield and N fertilizer. However, our findings are not directly comparable to those derived from Burke & Lobell (PNAS, 2017) because:

- (i) differences in yield source: the previous study used survey- and satellite-based yield measurements, while we used direct yield measurements derived from crop-cuts.
- (ii) difference in number of variables included in the analysis: the previous study did not consider the effect of management practices other than N and kg of hybrid seed, while we accounted for planting dates, plant density, seed type, NPK fertilizer rate and placement, weeding, and others.
- (iii) differences on the method to account for environmental variation: the previous study did not account for differences in biophysical background as determined by spatio-temporal variation in climate & soil, while we stratified fields based on climate zones and included soil, terrain, and in-season weather variables to account for residual environmental variation within climate zones.
- (iii) differences in ranges of independent variables: the previous study explored a relatively narrow range of management practices as the survey data came from farmers who followed OAF's technical packages, while our analysis included farmers following or not the OAF program, which helped to evaluate yield responses across wide ranges of agronomic variables
- (iv) number of fields: the previous study included 447 fields, whereas ours includes 14,773 fields.

Because our study is not comparable to that of Burke & Lobell (PNAS, 2017), there is no need to reevaluate the robustness and credibility of our results in view of the findings of this previous paper, and we also do not see the need to bring this issue into the discussion of the revised MS. Moreover, we note that our results do not contradict the results from this previous paper and the comment from Reviewer #1 focuss only on the differences in reported errors between studies – differences that can be attributed to the abovementioned differences between studies.

3. "State of art" methodology

The manuscript refers to the use of Gradient Boosting Machines (GBM) or Random Forest as “state of art” methodologies. While these methods are valuable, classifying them as the forefront of machine learning technology might be misleading. “state of art” often implies the use of the most advanced techniques available, which is not the case here. I am not suggesting a shift to more complex machine learning or deep learning algorithms, as the appropriateness of the method is paramount. However, the current portrayal in the manuscript might overstate the sophistication of the employed methods.

Response: We never referred to the use of GBM as a “state of art” methodology. Thus, no change was made to the manuscript in response to this comment.

4. Causal Analysis

The analysis in the paper seems to predominantly focus on correlation and predictive performance rather than causal inference. Given that the primary objective is to understand the drivers rather than merely develop predictive models, a transition to causal inference is essential. In fact, GBM is capable of contributing to causal inference, but this aspect has not been sufficiently explored in the current manuscript. Incorporating and elucidating a causal analysis approach would significantly enhance the depth and impact of the research.

Response: We appreciate reviewer’s suggestion and encouragement to include new analyses. While recent developments in the fields of machine learning and causal analysis are intriguing, we are not aware of any study establishing causal relationships from observational data in the field of agriculture. Even if we applied such an approach, the causal relations we might find might be challenged by others for being derived from observations rather than experiments so we prefer not to include new analysis for which a theoretical/empirical basis is lacking. On the other hand, our study is based on +10,000 observations coupled with soil and weather databases and spatial frameworks to account for environmental variation, which we believe make our findings robust regardless the lack of a formal experimental design. We added text elaborating on the comment by Reviewer #1 in the revised MS (L 234-238 of revised MS).

Comments from Reviewer #3:

I have gone through the paper and the authors’ response to the reviewers’ comments, specifically to reviewer #2. I agree that the paper used valuable dataset collected from African smallholder farms and the analysis of such data would add value to inform the global community at large and the agricultural literature specifically.

In their response, the authors have tried to clarify and/or provide additional information and data/analysis to the issues raised by the reviewers and this has improved the quality and relevance of the paper. However, there are outstanding issues that the authors didn’t address fully, or their responses raised additional questions or concerns, particularly to the issues raised by reviewer #2.

Response: we appreciate the positive feedback from Reviewer #3 about the revised MS and support for publication after addressing remaining concerns. See our detailed responses below

There are two major concerns that need additional clarification from the authors.

1. In their response to reviewer #2's concern on missing soil parameters, the authors stated "... as requested, we included other soil properties in our analysis as covariables, such as soil organic carbon, pH, CEC, and clay content, and we showed that the range of soil properties explored by our database was similar to that across the whole maize producing area in SSA (Supplementary Fig. S1). Inclusion of the new variables into our regression tree analysis (and new machine learning analysis) did not modify the main findings from our original MS and did not improve substantially the explanatory power of our models".

This statement raises a major concern on the sensitivity of the model/methodology used for analysis. In a region with diverse terrain, soil type and soil nutrient conditions, it is hardly possible to believe that carbon, pH, CEC and soil texture parameters do not affect a given analysis across the region. To support this claim, the authors need to show evidence that their method/model is sensitive to input parameters.

Response: See our previous response to Reviewer #1 in relation to stratification of the fields. Briefly, we stratified fields by agro-climatic zones to account for environmental gradients over larger region, which considerably narrowed the ranges in soil and climate variables within each of them, and subsequently performed a separate analysis to identify management factors with greatest influence on crop yield for each climate zone, which also included field-specific soil, terrain, and in-season weather variables to account for any residual variation in these variables.

The fact that soil variables did not appear too frequent in our regression trees performed at CZ level can be attributed to (i) relatively small variation in soil properties within CZ, (ii) larger impact of agronomic practices, terrain, and in-season weather on yield compared with soil properties, or (iii) a combination of both. Another possibility is that current soil databases are not sufficiently granular and accurate for the purpose of field-level studies – there is little we can do about this. Whatever the explanation is, it is not likely to influence the main findings from our study – indeed, it is remarkable that despite this potential source of uncertainty, we are still able to find strong and clear effects of management practices on yields. We added text to the revised MS to clarify on the stratification methods and also to elaborate on the findings about soils (L 81-83 & 470-480 of revised MS).

2. There is no lack of evidence on the role of good agronomic practices in increasing productivity in SSA, and the findings from this paper are not new. The authors' response on the concern raised by reviewer #2 regarding the full adoption assumption of the identified agronomic practices to project future production levels in SSA is not convincing. Although the potential to increase maize productivity through increasing the level of inorganic fertilizers, use of hybrid seeds and proactive crop protection practices is clear to smallholders farmers and extension workers, the major challenge is that most smallholders do not have the required financial resources, credit services,

agricultural insurance and market access to apply the inputs and hence the low adoption of improved seeds, higher fertilizer rates and use of agro-chemicals across the region. The lack of consideration of these adoption constraints led the authors to project a doubling of maize productivity in SSA, which is unrealistic. I think the authors could consider different adoption levels (which could vary among sub-regions) in their future production projection so that their estimates come closer to “reality”. This also provide readers, particularly those in the region, to consider what is possible/feasible based on their knowledge and experience.

Response: This point is well taken. We respectfully note that we are not trying to forecast what will happen in SSA but rather report on the outcomes of ‘what-if’ type of scenarios, in which there is a conscious and unprecedented investment on AR+D programs, and associated policy and institutions, to ensure smallholders in SSA access to basic agronomic inputs and adoption of better agronomic practices. We prefer to simply show what would happen following the historical trajectory of yield (‘business as usual’ scenario) and an intensification scenario in which a package of basic good agronomic practices is fully adopted across the region over the next 30 years. If adoption is less than 100%, it means the outcomes will fall somewhere between the two scenarios and that’s easy to estimate based on the values shown in Table 1. For example, if one would like to estimate the outcomes derived from a 50-% adoption scenario, it only requires to average the outcomes of the BAU and intensification scenarios – we feel there is no need to add an extra row and additional text for such simple calculation. In any case, failure to intensify at scale would only add more pressure on maize imports and/or land requirements and that is the main message the paper want to deliver. We agree with reviewer #3 that lack of access to financial resources, credit services, and agricultural insurance represents a major challenge for adoption of better management practices in SSA. We added text to the revised MS to highlight these challenges and the need for strong policy interventions (L 314-317 of revised MS).

3. Since the authors claim a doubling of maize productivity in SSA with the use of agronomic practices they identified (which could motivate donors and governments), reader would want to know the cost of full or partial implementation of the recommendations. It would make the paper useful and informative, if the authors could give an estimation of the investment required to adopt the practices, and discuss it relative to on-going investments in the region and the gap that needed to be filled.

Response: We added text to the revised MS to report on the costs of implementing these practices in SSA and discussed their implications for farmers and other stakeholders (L 320-325 of revised MS).

Reviewers' Comments:

Reviewer #1:

Remarks to the Author:

Some of my previous suggestions have been considered in this revision by adding discussions. While I still have reservations regarding the analysis, I also recognize the significance of the findings derived from such an extensive field dataset and the value of communicating this study sooner than later. Therefore, I am ok with accepting this paper should other reviewers agree as well.

Reviewer #3:

Remarks to the Author:

I have gone through the authors response and I am glad that the authors tried to address the issues raised one by one. They have also incorporated text in the manuscript to give context to readers. However, the responses didn't address the issues raised directly, and this causes concern. For examples, the authors responded to the concern raised on importance of soil parameters and model sensitivity as follows:

"The fact that soil variables did not appear too frequent in our regression trees performed at CZ level can be attributed to (i) relatively small variation in soil properties within CZ, (ii) larger impact of agronomic practices, terrain, and in-season weather on yield compared with soil properties, or (iii) a combination of both".

These are possible reasons but without evidence - the authors could support these claims with an analysis of the soil data they used in their model. Moreover, the authors response doesn't address the issue of model sensitivity with empirical evidence.

How was the cost mentioned in L322-324 estimated?

Responses to Reviewers' Comments (and our responses in red)

We thank the editor and the two reviewers for their comments and support for publication. We have addressed their comments in the revised MS. Our responses are shown below in red.

Comments from Reviewer #1:

Some of my previous suggestions have been considered in this revision by adding discussions. While I still have reservations regarding the analysis, I also recognize the significance of the findings derived from such an extensive field dataset and the value of communicating this study sooner than later. Therefore, I am ok with accepting this paper should other reviewers agree as well.

Response: thanks.

Comments from Reviewer #3:

I have gone through the authors response and I am glad that the authors tried to address the issues raised one by one. They have also incorporated text in the manuscript to give context to readers.

Response: thanks.

However, the responses didn't address the issues raised directly, and this causes concern. For examples, the authors responded to the concern raised on importance of soil parameters and model sensitivity as follows: "The fact that soil variables did not appear too frequent in our regression trees performed at CZ level can be attributed to (i) relatively small variation in soil properties within CZ, (ii) larger impact of agronomic practices, terrain, and in-season weather on yield compared with soil properties, or (iii) a combination of both". These are possible reasons but without evidence - the authors could support these claims with an analysis of the soil data they used in their model. Moreover, the authors response doesn't address the issue of model sensitivity with empirical evidence.

Response: This is well taken. We would like to highlight that the goal of our study is NOT to determine the exact contribution of management, weather, and soil at explaining field-to-field yield variation within a climate zone (CZ), but rather identifying management factors with greatest influence on farmer yields for a given CZ. We included soil and climate factors into the analysis to facilitate the identification of these management factors – again, the purpose was not to analyze which source of variation (management, climate, soil) accounted for most of the variation in farmer yields. We have added text to clarify the focus of our study and avoid confusion (L 487-493 of revised MS).

In response to the specific comments made by Reviewer #2, we have added two new figures showing empirical evidence that helps explain why the statistical models did not identify soil properties as a major driver for yield variation within CZs. These figures show that (i) for most soil properties, variation within CZ is smaller compared with variation among CZs (**new Supplementary Fig. S8 in revised MS**) and (ii) within CZs, management practices have a larger impact on farmer yield compared with soil, terrain, and climate factors (**new Supplementary Fig. S9 in revised MS**). Furthermore, we note that the effect of soil properties on yield is partially confounded by climate and topography given the correlations among these factors (e.g., elevation, temperature, and SOM) so it is not really possible to parse out the yield variation that is attributable to each factor (**Supplementary Fig. S4**). We have added text elaborating on these findings (L 476-492 of revised MS).

How was the cost mentioned in L322-324 estimated?

Response: As requested, we added text explaining how the cost was estimated (L 534–541 of revised MS).